# VL-Reader: Vision and Language Reconstructor is an Effective Scene Text Recognizer

## ABSTRACT

Text recognition is an inherent integration of vision and language, encompassing the visual texture in stroke patterns and the semantic context among the character sequences. Towards advanced text recognition, there are three key challenges: (1) an encoder capable of representing the visual and semantic distributions; (2) a decoder that ensures the alignment between vision and semantics; and (3) consistency in the framework during pre-training, if exist, and fine-tuning. Inspired by masked autoencoding, a successful pre-training strategy in both vision and language, we propose an innovative scene text recognition approach, named VL-Reader. The novelty of the VL-Reader lies in the pervasive interplay between vision and language throughout the entire process. Concretely, we first introduce a Masked Visual-Linguistic Reconstruction (MVLR) objective, which aims at simultaneously modeling visual and linguistic information. Then, we design a Masked Visual-Linguistic Decoder (MVLD) to further leverage masked vision-language context and achieve bi-modal feature interaction. The architecture of VL-Reader maintains consistency from pre-training to fine-tuning. In the pre-training stage, VL-Reader reconstructs both masked visual and text tokens, while in the fine-tuning stage, the network degrades to reconstruct all characters from an image without any masked regions. VL-reader achieves an average accuracy of 97.1% on six typical datasets, surpassing the SOTA by 1.1%. The improvement was even more significant on challenging datasets. The results demonstrate that vision and language reconstructor can serve as an effective scene text recognizer.

## CCS CONCEPTS

• **Applied computing → Optical character recognition**.

## KEYWORDS

Scene Text Recognition, OCR, Vision-Language Reconstruction

## 1 INTRODUCTION

Reading text from natural scenes has drawn significant attention in recent years since it is a crucial prerequisite for numerous computer vision tasks, including scene understanding, autonomous driving, and document-based large language models. Scene text recognition (STR), as an essential component in scene text reading, aims to decode a natural scene text image into a sequence of characters.

*ACM MM, 2024, Melbourne, Australia*
© 2024 Copyright held by the owner/author(s). Publication rights licensed to ACM.
ACM ISBN 978-x-xxxx-xxxx-x/YY/MM
https://doi.org/10.1145/nnnnnnn.nnnnnnn

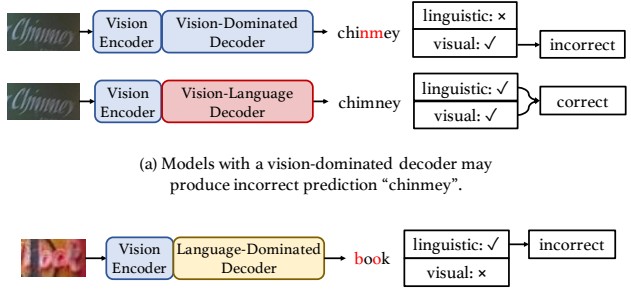

(a) Models with a vision-dominated decoder may produce incorrect prediction "chinmey".

(b) Models with a language-dominated decoder may produce incorrect prediction "book".

**Figure 1: (a) Models with vision-dominated decoders mainly rely on visual context and are incapable of handling low-quality images. (b) Models with language-dominated decoders mainly rely on linguistic context and may generate semantically correct but visually incorrect predictions.**

Retrospective studies show a steady stream of methods propelling the advancement of STR, including vision-dominated methods [16, 34, 35], and language-aware methods [4, 9].

A comprehensive consideration of both vision and language is essential in designing advanced STR. Since text images possess both the textual texture in stroke patterns and the semantics in words or lines. The significance of the interplay between vision and semantics becomes evident when dealing with occluded characters, blurred backgrounds, or messy handwriting. Previous vision-dominated methods [16, 34, 47] treat characters simply as distinct visual symbols and directly classify them into different categories mainly based on visual features, overlooking the underlying semantics. Recent language-aware methods [44, 46, 49] incorporate a language-aware module into the decoding stage to rectify recognition results, but fail to adequately account for the collaborative influence of vision and semantics. Both of these approaches fail to simultaneously consider bi-modal information in both the encoding and decoding stages. Vision-dominated decoder tends to prioritize visual facts, which is occasionally ambiguous, as exemplified by the blurred characters "m" and "n" in the Fig. 1(a) representing "chimney". While language-dominated decoder is more likely to select a word with a higher frequency in the dictionary, such as "book" instead of "rock" as seen in Fig. 1(b).

To enhance robustness against various visual conditions and improve the model's understanding of context and syntax in text, we advocate that an optimal text recognition model should possess the following three key properties: 1) representativeness: an encoder capable of representing the visual and semantic distribution, 2) multi-modal decoding ability: a decoder that ensures the

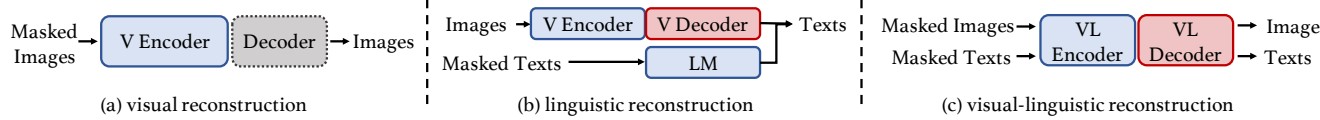

**Figure 2: The comparison between different reconstruction pipelines. (a) Visual Reconstruction follows the pipeline of MAE and its decoder will be discarded in the recognizer (dashed gray box). (b) Linguistic Reconstruction utilizes a standalone language model for linguistic refinement after visual results. (c) Our Visual-Linguistic Reconstruction reconstructs both visual and linguistic information and can inherit the entire architecture.**

alignment between vision and semantics, and 3) structural consistency: consistency in the framework during the pre-training and fine-tuning. In terms of representativeness, masked autoencoding has been shown to be effective in learning either vision ( MAE [11] ) or language ( BERT [19] ) representation. In this work, we investigate that text recognition itself can be viewed as the reconstruction of masked characters. Therefore, we can seamlessly transfer a vision-language reconstruction model into a text recognition model, requiring no extra layers. This results in a simple yet effective text recognizer, which we have named VL-Reader. Concretely, we first introduce a Masked Visual-Linguistic Reconstruction (MVLR) objective, which learns visual and semantic representations by means of self-supervised masked autoencoders. Then, we devise a Masked Visual-Linguistic Decoder (MVLD) to further leverage masked vision-language context and achieve bi-modal feature interaction. The architecture of VL-Reader maintains consistency from pre-training to fine-tuning. In the first training stage, the VL-Reader reconstructs both masked visual and text tokens, while in the second stage, the network degrades to reconstruct all characters from an image without any masked regions.

Different from previous methods that reconstruct signals from either visual or linguistic modality, our work emphasizes the importance of jointly reconstructing both visual *and* linguistic signals (see Fig. 2). The proposed training objective MVLR forces effective cooperation between the visual and linguistic modalities, thereby aiding in building a strong cross-modal feature representation. Thanks to the generality of Transformer [40], we are able to model visual and textual information, which have different levels of densities, in the same dimension. Furthermore, the entire architecture can be seamlessly transitioned from pre-training to fine-tuning, simply with a change of masking matrix, which will be detailed in Sec. 3.2.

Experiments are conducted on six standard benchmarks as well as seven more challenging benchmarks. VL-Reader achieves state-of-the-art performance on all thirteen benchmarks (Table 1 and Table 2). The magnitude of improvement achieved is particularly significant when compared to the already high baseline accuracy in the field. Moreover, our method demonstrates an even more significant performance boost on seven more challenging datasets, additionally validating the effectiveness and robustness of our proposed VL-Reader.

The contributions of our work are summarized as follows:

- We propose VL-Reader, a novel STR approach that leverages masked vision and language for auto-encoding and reconstruction for text decoding. This method demonstrates a concise yet highly effective architecture.

- We introduce masked visual-linguistic reconstruction for STR, which jointly learns representations of the vision and semantics of text images.
- We design a cross-modal masked visual-linguistic decoder, which serves the dual purpose of supervising the reconstruction task and acting as the output for recognition results.
- Experiments on extensive benchmarks show that the proposed VL-Reader outperforms the existing methods by a significant margin, demonstrating that vision and language reconstructor can serve as an effective scene text recognizer.

## 2 RELATED WORK

### 2.1 Vision-dominated Methods

Vision-dominated methods treat characters as distinct visual symbols and directly classify them into different categories solely based on visual features. CTC-based methods [12, 13, 34] utilize a CTC decoder for converting a sequence of features into a sequence of characters. Segmentation-based methods [24, 25, 42] employ a semantic segmentation pipeline to address the scene text recognition task. Recent methods [16, 47] integrate masked image modeling (MIM) as an additional pre-training step to enhance visual representation. DiG [47] combines masked image modeling and contrastive learning to improve discriminative and generative representation. MAE-Rec [16] adopts the pipeline of MAE and utilizes large-scale unlabeled data to develop a strong visual encoder. Although these approaches have achieved promising progress in standard benchmarks, their lack of integration of linguistic knowledge may hinder performance when handling low-quality images.

### 2.2 Language-aware Methods

Recent approaches [4, 9, 44, 46, 49] have recognized the importance of linguistic knowledge and have started to integrate it into their systems. SRN [49] adopts a semantic reasoning module to model contextual information and achieves promising results. ABINet [9] introduces an iterative refinement stage, where linguistic knowledge is used to progressively correct text recognition results with a standalone language model. VisionLAN [46] successfully integrates visual and linguistic information into a single model. MGP-STR [44] recognizes characters in a multi-granularity approach by additionally predicting subwords. PARSeq [4] utilizes linguistic knowledge in an implicit way by using a permuted auto-regressive sequence model. Despite integrating linguistic knowledge into their models, these methods typically consider it as merely a supplement to visual knowledge. Thus we need a comprehensive exploration of jointly modeling visual and linguistic knowledge.

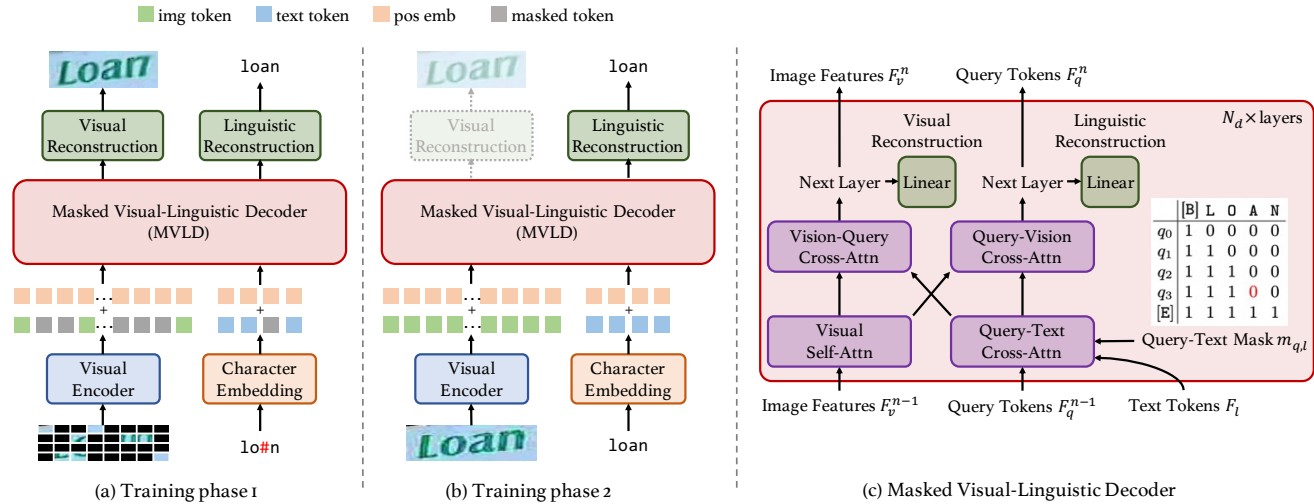

**Figure 3: Overall architecture of VL-Reader and its detailed structure for the masked visual-linguistic decoder. In the first training phase, VL-Reader is trained under the supervision of MVLR. In the second phase, VL-Reader disables the visual reconstruction task and focuses on the text recognition task only. Black patches indicate masked visual patches and "#" indicates a masked language token.**

**Figure 4: The generation process of query-text attention mask $m_{q,l}$. In the second training phase, the attention mask for masking out characters will be a matrix completely filled with "1"s.**

## 3 METHODOLOGY

### 3.1 Overall Architecture

The overview of the proposed VL-Reader is depicted in Fig. 3(a) which consists of a visual encoder, a linguistic embedding layer, a masked visual-linguistic decoder, and two reconstruction heads. We will first introduce the proposed training objective Masked Visual-Linguistic Reconstruction (MVLR). After that, the details of all components are presented in the subsequent sections.

### 3.2 Masked Vision-Language Reconstruction

In recent years, drawing inspiration from MAE [11], several approaches [16, 47] have adopted visual reconstruction as the training objective to enhance visual representation learning. These methods attempt to reconstruct masked visual patches based on the unmasked ones. By deciphering the underlying information contained within unmasked visual patches, they can forge a strong visual representation.

However, models that solely reconstruct visual information still face several challenges. (1) Visual reconstruction is trained solely based on visual information and lacks integration of linguistic information. Models trained with such an objective may have trouble when handling low-quality images. (2) After the visual reconstruction training, only the encoder component can be utilized for fine-tuning text recognizers while the decoder part will be wasted.

In this work, we propose a novel training objective termed Masked Visual-Linguistic Reconstruction (MVLR) for robust scene text recognition. MVLR is designed to (1) simultaneously reconstruct visual and linguistic information and (2) inherit the entire architecture across different training stages.

**Visual-Linguistic Reconstruction Objective.** The reconstruction objective can be further divided into two parts, the visual reconstruction and the linguistic reconstruction. In the case of visual reconstruction, the model is trained to reconstruct the masked visual patches utilizing both the unmasked patches *and* unmasked text tokens. Similarly, for linguistic reconstruction, the model is trained to reconstruct the masked text tokens by leveraging both the unmasked tokens *and* unmasked visual patches. By effectively integrating visual and linguistic information, VL-Reader is capable of learning strong cross-modal feature representation.

**Visual-Linguistic Masking.** For an image $I \in R^{H \times W \times C}$ and its corresponding text sequence $T$, we first cut the image with patch size $(p_h, p_w)$ and then randomly mask a subset of patches with ratio $r_v$. The remaining patches $I^*$ are visible patches and will be encoded by the ViT encoder. For text sequence $T$, we randomly

mask a subset of tokens with ratio $r_l$ and replace such tokens with text mask token $[\text{MASK}_1]$. All the text tokens will be encoded by a text embedding vector, which follows PARSeq [4]. The effects of $r_v$ and $r_l$ will be discussed in the ablation studies.

**Encoder.** Our encoder consists of a visual encoder to encode visual information and a linguistic encoder to convert character tokens into embedding. Following prior works [4, 16, 44], the visual encoder employs the architecture of ViT [8] but is applied only on visible patches $I^*$. The visual encoding process can be formulated as follows:

$$F_v = ViT(i|i \in I^*) + Vec(i|i \notin I^*), \quad (1)$$

where $i$ is the image patch. The first part denotes the visible patches encoded by $ViT$, and the second part denotes the masked patches with random vectors $Vec$. $F_v$ is the final visual embedding.

As for encoding context information, we adopt a text embedding layer to convert character tokens into context embedding. The contextual encoding process can be formulated as follows:

$$F_l = Emb(t|t \in T), \quad (2)$$

where $t$ is the character token. Different from visual encoding, the visible characters and masked characters are encoded in the same way. $F_l$ represents the final contextual embedding.

Note that, both visual and text mask token $[\text{MASK}_v]$ and $[\text{MASK}_1]$ are shared and learnable vectors, indicating a visual patch or a text token to be reconstructed. Besides, we omit position embedding in the formulas for brevity, both visual embedding and contextual embedding contain position embedding.

**Masked Visual-Linguistic Decoder.** The Masked Visual-Linguistic Decoder (MVLD) has $N_d$ layers. The detailed structure of each layer is illustrated in Fig. 3 (c). During decoding, we use a sequence of query tokens $F_q \in R^{L_q \times C}$ as the bridge to gather visual-linguistic information and carry out cross-modal feature interaction. Concretely, we adopt a visual self-attention to model visual context, a query-language cross-attention to model linguistic context, and a visual-linguistic cross-attention to model the interaction between visual and linguistic modality. All attention layers are implemented with standard Multi-Head Attention $\text{MHA}(\mathbf{q}, \mathbf{k}, \mathbf{v}, \mathbf{m})$, where $\mathbf{q}$, $\mathbf{k}$, $\mathbf{v}$ and $\mathbf{m}$ indicates *query*, *key*, *value* and an optional attention *mask*. The decoding process in $n$ th layer can be formulated as follows:

$$H_v = \text{MHA}(F_v^{n-1}, F_v^{n-1}, F_v^{n-1}) \quad (3)$$

$$H_q = \text{MHA}(F_q^{n-1}, F_l, F_l, m_{q,l}) \quad (4)$$

$$F_v^n = \text{MHA}(H_v, H_q, H_q) \quad (5)$$

$$F_q^n = \text{MHA}(H_q, H_v, H_v) \quad (6)$$

where, $n \in [0, 1, \ldots, N_d]$ indicates the $n$ th layer of MVLD, $H_v$ and $H_q$ represent hidden layers for visual self-attention and linguistic cross attention respectively. $F_v^{n-1}$ is the output visual features of the $(n-1)$ th layer and is also the input of the $n$ th layer. $F_l$ is the encoded linguistic feature introduced in Equation 2, and remains the same across all $N_d$ layers. $F_q^{n-1}$ is the input query tokens of the $n$ th layer and $F_q^0$ is randomly initialized before the first layer. We employ attention mask $m_{q,l}$ in the query-text cross-attention to avoid information leakage. The positions with masked characters are set as $[\text{-inf}]$ in $m_{q,l}$, as exemplified in Fig. 4. Concretely, we adopt the permuted attention mask for robust context modeling and the masked attention mask to avoid information leakage. Then

we merge them with an AND operation to form our query-text attention mask $m_{q,l}$.

For the reconstruction of each layer, we also send the decoded visual and linguistic features $F_v^k$ and $F_q^k$ to the visual and linguistic reconstruction head respectively:

$$v^n = Head_v(F_v^n) \quad (7)$$

$$l^n = Head_l(F_l^n) \quad (8)$$

where $v^n$ and $l^n$ indicate the reconstructed visual patches and linguistic tokens of the $n$ th layer. $Head_v$ and $Head_l$ both consist of several linear layers. Mean Square Error (MSE) and CrossEntropy loss functions are adopted to supervise the visual and linguistic reconstruction objectives respectively.

**Optimization.** The model is trained end-to-end using the following objective:

$$L = \lambda_v L_v + \lambda_t L_l \quad (9)$$

where $L_v$ and $L_l$ indicate the loss of visual and linguistic reconstruction respectively.

We use Mean Square Error (MSE) for $L_v$:

$$L_v = \frac{1}{|M_v|} \frac{1}{N_d} \sum_{n=1}^{N_d} \sum_{i \in M_v} (v_i^n - y_i)^2 \quad (10)$$

where $M_v$ represents the masked visual patches, $v_i^n$ and $y_i$ represents the $i$ th reconstructed and ground-truth pixel of the $n$ th layer.

We use Cross-Entropy loss for $L_l$:

$$L_l = \frac{1}{|M_l|} \frac{1}{N_d} \sum_{n=1}^{N_d} \sum_{i \in M_l} L_{ce}(l_i^n, t_i) \quad (11)$$

where $M_l$ indicates the masked language tokens, $l_i^n$ and $t_i$ indicates the $i$ th reconstructed and ground-truth language token of the $n$ th layer.

## 3.3 Training and Inference

**Training.** The training process of VL-Reader has two phases, as illustrated in Fig. 3. In the first phase, we adopt MVLR as the training objective, in which both image and text reconstruction are implemented. Under the guidance of MVLR, our model learns a cross-modal representation. Compared to the methods that use a single visual reconstruction, our method takes into consideration the accuracy of text during the visual reconstruction process. On the contrary, compared to the method of simply using a language model for text correction, our approach also takes into account the visual context during text reconstruction.

The second training phase, also known as the fine-tuning phase, involves performing text recognition tasks based on the architecture from the first phase. In this stage, we deactivate the visual reconstruction and focus solely on the linguistic reconstruction.

Specifically, we set $\lambda_v$ and $\lambda_l$ as 1.0, $r_v$ as 0.75, and $r_l$ as 0.2 during the first training phase to activate both visual and linguistic reconstruction. We set $\lambda_v$ as 0 during the second training phase to deactivate visual reconstruction and focus on linguistic reconstruction. To further enhance the language capabilities during the recognition stage, we utilize the permutation language model strategy from PARSeq [4]. This only requires substituting the attention

mask in the MVLD module (see Fig. 4). In summary, the two training phases have the same model architecture. The only variation lies in the presence of the mask in the input and the attention mask in the decoder.

**Inference.** We adopt an auto-regressive pattern to decode characters in the reading order during inference. For the first iteration, we use only the first query token $\mathbf{q}_1$ of the initial query sequence $F_q^0$. And for the succeeding iteration $t$, we use query tokens $[\mathbf{q}_1, \mathbf{q}_2, \ldots, \mathbf{q}_t]$. We set the visual masking ratio $r_v$ as 0 to allow full visual perception. Inspired by prior works [4, 9], a refinement stage is employed to further adjust predicted results. We use the cloze mask as the attention mask $m_{q,l}$ for query-text cross-attention in the refinement stage.

## 4 EXPERIMENTS

### 4.1 Datasets and Implementation Details

**Datasets.** Following prior works [4][16], we use both synthetic and real datasets for training. The synthetic datasets include MJSynth (MJ) [14, 15] and SynthText (ST) [10]. The real datasets are introduced in PARSeq [4], including ArT [5], COCO-Text [41], LSVT [39], MLT-19 [29], RCTW17 [36], ReCTS [52], Uber [53], TextOCR [37] and OpenVINO [22]. To evaluate our method as fair as possible, we train our model on three groups of datasets (*i.e.*, synthetic datasets only, real datasets only and a mixture of synthetic and real datasets).

To conduct a fair comparison with previous methods, we follow the evaluation protocol of PARSeq [4]. Concretely, (1) six standard benchmark datasets including IIIT5K (IIIT) [27], ICDAR2013 (IC13) [18], ICDAR2015 (IC15) [17], Street View Text (SVT) [43], Street View Text-Perspective (SVTP) [31] and CUTE80 (CUTE) [33] are used for evaluation. (2) In addition, we also evaluate our model on seven more challenging datasets, including two occluded datasets WOST and HOST [46], two handwritten datasets IAM [26] and CVL [21] and three large-scale datasets COCO-Text [41] (low-resolution, occluded), ArT [5] (curved, rotated) and Uber [53] (vertical, rotated) to validate the robustness of our methods in more challenging scenarios.

**Implementation Details.** The training process has two phases. In the first phase, we employ MVLR as our training objective to simultaneously reconstruct visual and linguistic information. In the second phase, we set $\lambda_v, r_v, r_l$ as 0 to disable visual reconstruction and focus on the text recognition task only. We train our model for 20 epochs for real datasets or 10 epochs for synthetic datasets with an initial learning rate of $7e - 4$ in the first phase. We fine-tune our model for another 10 epochs with an initial learning rate of $1e - 4$ in the second phase. All models are trained with a total batch size of 768 on 4 GPUs (192 images per GPU).

Following previous state-of-the-arts [2, 44, 47], we use ViT-Base [8] as visual encoder with a patch size of $4 \times 8$. Unless specified, the decoder depth $N_d$ is set to 4, and the visual masking ratio $r_v$ and linguistic masking ratio $r_l$ are set as 0.75 and 0.2 respectively. We employ the Adam optimizer [20] together with the 1cycle [38] learning rate scheduler. During the second training phase, the Permutation Language Modeling (PLM) [48] introduced in [4] is also adopted for better context modeling and we set the number of permutations as 6. Following prior works [4, 35], the maximum label length is set to 25. Images with label lengths larger than 25 will be

neglected during training. During evaluation, we set the charset size as 36, including lower-case alphanumeric characters.

For image pre-processing, RandAugment [6] with 3 layers and a magnitude of 5 excluding Sharpness is employed as our data augmentation strategy. Following PARSeq [4], we also add Invert, GaussianBlur and PoissonNoise due to their effectiveness in STR task. After augmentation, all images will be resized to a fixed size of (32, 128) and will be normalized to $[-1, 1]$.

### 4.2 Comparisons with State-of-the-Arts

To make a fair comparison with prior arts, We follow the evaluation protocol of PARSeq [4] and choose word accuracy as our evaluation metric. We evaluate our model on thirteen benchmark datasets, including six standard benchmarks and seven more challenging benchmarks.

*4.2.1 Standard Benchmarks.* We evaluate VL-Reader on six standard benchmark datasets, including three regular datasets (IIIT5K, SVT, and IC13) and three irregular datasets (IC15, SVTP, and CUTE80). Results are presented in Table 1. When trained on synthetic datasets, VL-Reader achieves state-of-the-art performance. Specifically, in comparison to previous SOTA methods, VL-Reader exhibits superior results on IIIT5K (97.1%), SVTP (92.9%), and CUTE80 (92.7%), while maintaining competitive results on the remaining datasets. The utilization of real-world data further amplifies the performance enhancement for the VL-Reader model, enabling it to attain state-of-the-art performance consistently across six standard benchmarks with an average accuracy of 96.9%. Furthermore, leveraging a hybrid dataset composed of both synthetic and real images, the VL-Reader model successfully maintains consistent performance improvements, ultimately attaining an average accuracy of 97.1% and setting new state-of-the-art benchmarks.

*4.2.2 More Challenging Benchmarks.* The model is also evaluated on seven more challenging benchmarks, including two occluded datasets (WOST and HOST), two handwritten datasets (IAM and CVL), and three large-scale datasets (COCO, ArT, and Uber). As indicated in Table 2, VL-Reader significantly outperforms previous works on the occluded datasets WOST(+4.4%) and HOST (+7.8%). One key factor may be that VL-Reader develops a robust cross-modal representation, enabling it to effectively handle visually challenging images. Furthermore, VL-Reader achieves a consistent performance boost across two handwritten datasets (IAM (+2.7%) CVL (+1.2%)) and three large-scale datasets (COCO (+2.4%), ArT (+0.7%) and Uber (+1.9%)). These results demonstrate the enhanced robustness of VL-Reader on large real world benchmarks.

By utilizing the hybrid dataset of synthetic and real images, VL-Reader consistently attains improved performance. However, VL-Reader does not demonstrate improved results on Uber. This might be attributed to the fact that synthetic images are mainly horizontal, which could hamper the performance on Uber as it primarily contains vertical and rotated images.

### 4.3 Ablation Study

*4.3.1 Effectiveness of MVLR.* The proposed MVLR plays a key role in the training process of VL-Reader. We conduct several experiments to validate the effectiveness and explore the impact of MVLR.

**Table 1: Word accuracy on six standard benchmark datasets. "S" represents synthetic datasets (MJ and ST), and "R" represents real datasets introduced by [4]. Superscript "−" and "+" represent using a subset of data and using external data respectively. The bold and underline results represent the best and the second-best respectively. † indicates results are from [4].**

| Methods | Train Data | Regular | | | | Irregular | | | | Weighted Avg. |
|---|---|---|---|---|---|---|---|---|---|---|
| | | IIIT | SVT | IC13 | | IC15 | | SVTP | CUTE | |
| | | 3000 | 647 | 857 | 1015 | 1811 | 2077 | 645 | 288 | |
| ESIR [51] | S | 93.3 | 90.2 | - | 91.3 | - | 76.9 | 79.6 | 83.3 | 86.8 |
| DAN [45] | S | 94.3 | 89.2 | - | 93.9 | - | 74.5 | 80.0 | 84.4 | 86.9 |
| RobustScanner [50] | S⁺ | 95.4 | 89.3 | - | 94.1 | - | 79.2 | 82.9 | 92.4 | 89.2 |
| TextScanner [42] | S | 93.9 | 90.1 | - | 92.9 | 79.4 | - | 84.3 | 83.3 | - |
| SRN [49] | S | 94.8 | 91.5 | 95.5 | - | 82.7 | - | 85.1 | 87.8 | - |
| VisionLAN [46] | S | 95.8 | 91.7 | 95.7 | - | 83.7 | - | 86.0 | 88.5 | - |
| TRBA [3] | S | 96.3 | 92.8 | 96.3 | 95.0 | 84.3 | 80.6 | 86.9 | 91.3 | 90.6 |
| ABINet [9] | S⁺ | 96.2 | 93.5 | 97.4 | - | 86.0 | - | 89.3 | 89.2 | - |
| ViTSTR-B [2] | S | 88.4 | 87.7 | 93.2 | 92.4 | 78.5 | 72.6 | 81.8 | 81.3 | 83.8 |
| PIMNet [32] | S | 95.2 | 91.2 | 95.2 | 93.4 | 83.5 | 81.0 | 84.3 | 84.4 | 89.5 |
| DiG-ViT-B [47] | S | 96.7 | 94.6 | - | **96.9** | 87.1 | - | 91.0 | 91.3 | - |
| TrOCR-Base [23] | S | 90.1 | 91.0 | 97.3 | 96.3 | 81.1 | 75.0 | 90.7 | 86.8 | 86.8 |
| MATRN [28] | S | 96.6 | 95.0 | **97.9** | 95.8 | 86.6 | 82.8 | 90.6 | **93.5** | 92.0 |
| MGP-STR [44] | S | 96.4 | **94.7** | 97.3 | 96.6 | **87.2** | **83.8** | 91.0 | 90.3 | 92.2 |
| LevOCR [7] | S | 96.6 | 92.9 | 96.9 | - | 86.4 | - | 88.1 | 91.7 | - |
| PARSeqₐ [4] | S | 97.0 | 93.6 | 97.0 | 96.2 | 86.5 | 82.9 | 88.9 | 92.2 | 91.9 |
| VL-Reader | S | **97.1** | 94.4 | 97.6 | 96.6 | 86.6 | 83.3 | **92.9** | 92.7 | **92.6** |
| CRNN† [34] | R | 94.6 | 90.7 | 94.1 | 94.5 | 82.0 | 78.5 | 80.6 | 89.1 | 88.5 |
| TRBA† [3] | R | 98.6 | 97.0 | 97.6 | 97.6 | 89.8 | 88.7 | 93.7 | 97.7 | 95.2 |
| ABINet† [9] | R | 98.6 | 97.8 | 98.0 | 97.8 | 90.2 | 88.5 | 93.9 | 97.7 | 95.3 |
| ViTSTR-S† [2] | R | 98.1 | 95.8 | 97.6 | 97.7 | 88.4 | 87.1 | 91.4 | 96.1 | 94.2 |
| PIMNet [32] | R⁻ | 96.7 | 94.7 | 96.6 | 95.4 | 88.7 | 85.9 | 88.2 | 92.7 | 92.6 |
| DiG-ViT-B [47] | R⁻ | 97.6 | 96.5 | - | 97.6 | 88.9 | - | 92.9 | 96.5 | - |
| PARSeqₐ [4] | R | 99.1 | 97.9 | 98.3 | 98.4 | 90.7 | 89.6 | 95.7 | 98.3 | 96.0 |
| MAE-Rec [16] | R⁺ | 98.5 | 97.8 | - | 98.1 | - | 89.5 | 94.4 | 98.6 | 95.6 |
| VL-Reader | R | 99.4 | **99.1** | 98.7 | 98.5 | **92.6** | **91.7** | 97.5 | **99.3** | 96.9 |
| VL-Reader | R+S | **99.6** | 98.5 | **99.4** | **99.3** | 92.4 | 91.4 | **98.1** | **99.3** | **97.1** |

**Table 2: Word accuracy on occluded, handwritten, and large-scale benchmark datasets. "S" represents synthetic datasets (MJ and ST), and "R" represents real datasets introduced by [4]. The bold and underline results represent the best and the second-best respectively. † indicates results are from [4].**

| Methods | Train Data | Occluded | | Handwritten | | Large-scale | | |
|---|---|---|---|---|---|---|---|---|
| | | WOST | HOST | IAM | CVL | COCO | ArT | Uber |
| | | 2416 | 2416 | 13752 | 12012 | 9825 | 35149 | 80551 |
| VisionLAN [46] | S | 70.3 | 50.3 | - | - | - | - | - |
| ViTSTR-S† [2] | S | - | - | - | - | 56.4 | 66.1 | 37.6 |
| DiG-ViT-B [47] | S | 82.3 | 74.9 | 87.0 | 91.3 | - | - | - |
| SeqCLR [1] | S | - | - | 79.9 | 77.8 | - | - | - |
| TextAdaIN [30] | S | - | - | 87.3 | 78.2 | - | - | - |
| CRNN† [34] | R | - | - | - | - | 66.8 | 62.2 | 51.0 |
| ViTSTR-S† [2] | R | 77.9 | 64.5 | - | - | 73.6 | 81.0 | 78.2 |
| ABINet† [9] | R | 85.0 | 72.2 | - | - | 76.5 | 81.2 | 71.2 |
| PARSeqₐ [4] | R | 85.4 | 74.4 | 89.7 | 90.0 | 79.8 | 84.5 | 84.1 |
| VL-Reader | R | 89.8 | 82.7 | **92.4** | **92.5** | 82.0 | 85.0 | **86.0** |
| VL-Reader | R+S | **92.9** | **87.3** | 92.0 | **92.5** | **82.2** | **85.2** | 84.7 |

**Table 3: Comparisons between enabling and disabling visual or linguistic reconstruction. "Knowledge" indicates integrating visual/linguistic knowledge or not.**

| Methods | Knowledge | | MVLR | Avg.(S) |
|---------|-----------|------------|------|---------|
| | Visual | Linguistic | | |
| #1 | √ | × | × | 94.7 |
| #2 | √ | √ | × | 96.1 |
| #3 | √ | √ | √ | **96.9** |

**Table 4: Comparisons of different model sizes. "Avg.(S) and Avg.(O)" represent the weighted average accuracy on six Standard and two Occluded benchmarks respectively.**

| Methods | Encoder | Avg.(S) | Avg.(O) | Params(M) |
|---------|---------|---------|---------|-----------|
| ABINet$_{LV}$ | - | 95.3 | 78.6 | 36.7 |
| VL-Reader | Tiny | 95.45 | 79.80 | 9.06 |
| VL-Reader | Small | 96.48 | 85.35 | 35.7 |
| VL-Reader | Base | 96.90 | 86.28 | 142 |

The baseline model (#1) bypasses the MVLR training phase and is directly trained solely on visual knowledge. In experiment (#2), linguistic knowledge is integrated alongside visual knowledge into our model, but the MVLR training phase is still bypassed. In Experiment (#3), we integrate visual and linguistic knowledge in the full setting.

The results can be seen in Table 3. By integrating linguistic knowledge, VL-Reader boosts recognition performance by +1.4%. With the addition of the proposed MVLR, there is an additional improvement of 0.8%, resulting in an overall improvement of 2.2%. Compared to experiment #2 which solely integrates linguistic information but is not trained under the MVLR objective, integrating MVLR could obtain a +0.8% performance gain, demonstrating the effectiveness of MVLR.

*4.3.2 Analysis of visual masking ratio $r_v$.* To examine the impact of varying visual masking ratios ($r_v$) on ultimate performance, we manipulated $r_v$ while maintaining a constant $r_l$ of 0.2 during the initial training phase. We fix the left hyper-parameters in various masking-ratio settings. The results are presented in Fig. 5 (a). The performance reaches the peak with a masking ratio around 0.7-0.75 and gradually decreases when enlarging or decreasing the visual masking ratio $r_v$. When $r_v$ is set to less than 0.65, the VL-Reader encounters a significant performance drop. The results demonstrate that the visual reconstruction task with an appropriate masking ratio is capable of boosting text recognition performance. As a result, we set the visual masking ratio $r_v$ as 0.75 in our work.

*4.3.3 Analysis of linguistic masking ratio $r_l$.* Similarly, we also examine the impact of varying linguistic mask ratios $r_l$. We set different $r_l$ while maintaining a fixed $r_v$ of 0.75. The results are shown in Fig. 5 (b). The performance of VL-Reader reaches the peak when $r_l$ is set around 0.15-0.2. However, further increasing the linguistic masking ratio results in a subsequent decrease in performance. This decline is attributed to the escalating training difficulty associated with enlarging $r_l$, as reconstructing 50% of characters based on the remaining 50% becomes increasingly challenging.

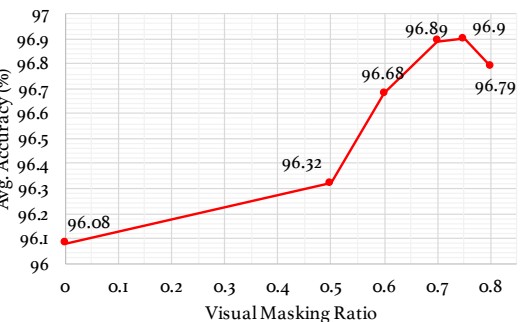

(a) Avg. Accuracy with different Visual Masking Ratio $r_v$

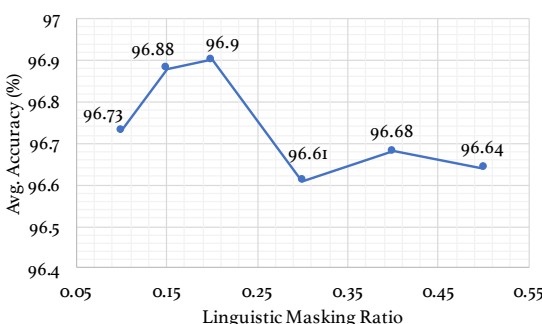

(b) Avg. Accuracy with different Linguistic Masking Ratio $r_l$

**Figure 5: Analysis of (a) different visual masking ratios $r_v$ and (b) different linguistic masking ratios $r_l$. All other parameters are fixed during training. VL-Reader reaches the highest average accuracy on six standard benchmarks around $r_v = 0.75$ and $r_l = 0.2$.**

*4.3.4 Analysis of Model Size.* We conduct an analysis of model size by implementing our model with different ViTs, specifically, tiny, small, and large. As can be seen from Table 4, when employing ViT-Tiny, VL-Reader can outperform ABINet by +0.15% on standard benchmarks and by +1.2% on occluded benchmarks with much smaller model size (9M *vs.* 36M). When utilizing ViT-Small, our VL-Reader consistently outperforms ABINet by +1.18% and +6.75% on standard and occluded benchmarks with similar model size. Furthermore, VL-Reader with ViT-Base achieves 96.9% and 86.28% on such benchmarks, surpassing all previous methods.

*4.3.5 Qualitative Results of Recognition.* We perform qualitative comparisons of VL-Reader and previous state-of-the-art models on typical images from standard and more challenging benchmarks. As illustrated in Fig. 6, we present some representative images to study the reason that VL-Reader succeeds but prior works fail. The results demonstrate that VL-Reader can obtain correct results on various challenging scenarios including occluded, artistic, blurred, and rotated images. Moreover, the VL-Reader is more robust to low-quality images and disturbances due to its comprehensive utilization of visual-linguistic context. For the occluded image "rock" (the second sample of "Occluded Text"), VL-Reader produces the correct answer while previous methods predict it as "book". In the heavily blurred scenario (refer to the "mandarin" sample, the fourth

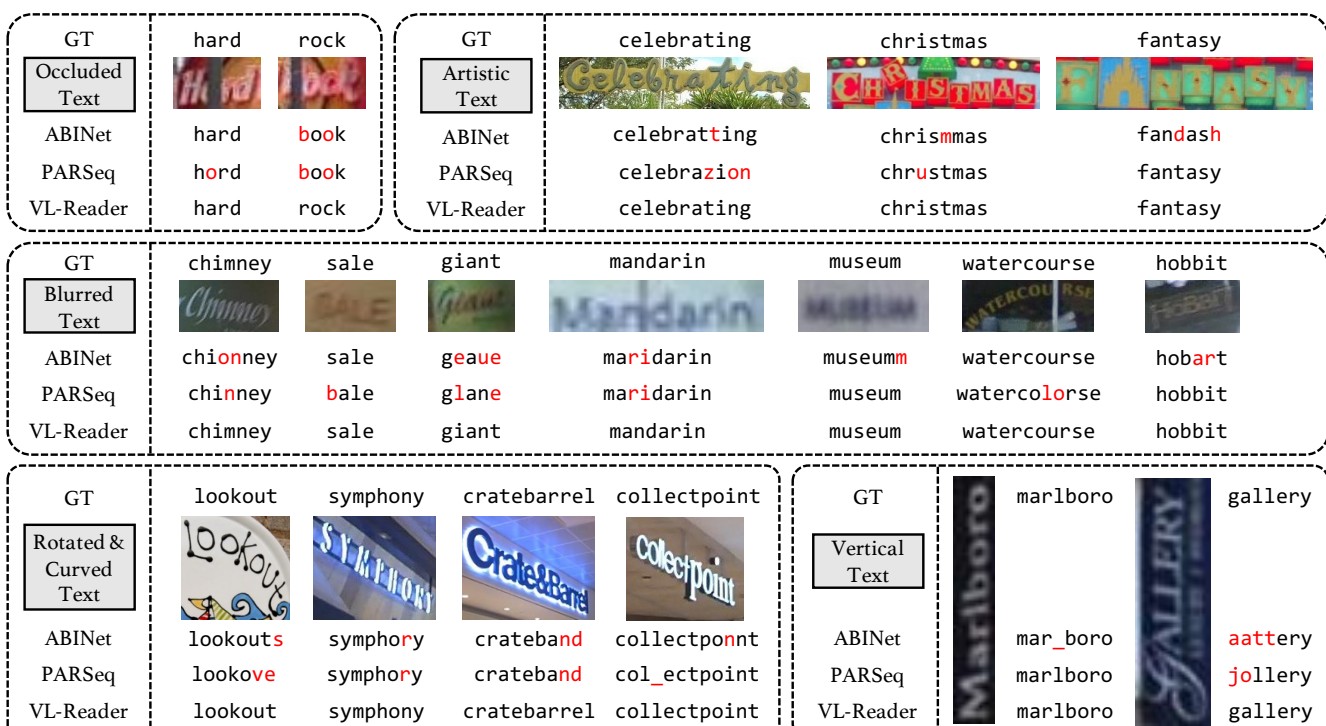

Figure 6: Qualitative comparison of VL-Reader and previous SOTA methods on challenging samples. Text from top to bottom are ground-truth text (top) and predicted results from ABINet(row#2), PARSeq (row#3) and VL-Reader (bot). Red characters indicate incorrect, missing, or redundant predictions.

instance in the "Blurred Text"), other methods incorrectly identify "n" as "ri", while VL-Reader recognizes it accurately. For the rotated scenario (see "symphony", the second sample of "Rotated & Curved Text"), the character "N" is hard to be distinguished from character "R" from the visual perspective due to adhered strokes under a perspective view. Thus Previous methods all incorrectly predict the character "N" as "R", while VL-Reader correctly identifies the word "symphony" with a valid linguistic meaning. These results demonstrate the robustness of VL-Reader on various challenging scenarios.

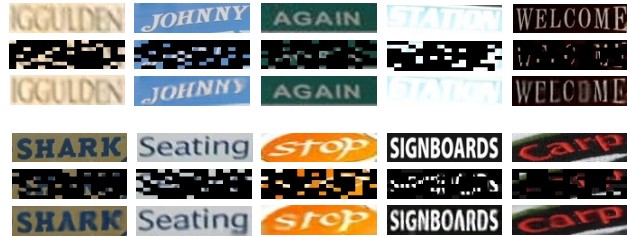

Figure 7: Reconstruction results on images of benchmark datasets (not used in training). For each column, we show the ground truth (top), the masked image (middle), and the reconstructed image (bottom). The visual masking ratio $r_v$ is set to 0.75.

*4.3.6 Reconstruction results of VL-Reader.* VL-Reader can simultaneously reconstruct visual and linguistic information. We showcase several of the reconstruction outcomes on images from benchmark datasets (i.e., images not utilized in training). As depicted in Fig. 7, VL-Reader effectively reconstructs most of the visual information even when the source image is heavily blurred (column#1). Additionally, the VL-Reader is able to reconstruct a character that is completely masked (the last character "g" in column#6). These reconstruction results confirm the successful acquisition of visual-linguistic representation by VL-Reader. Moreover, we also present a further discussion regarding the vision-language reconstruction results in our supplementary materials.

## 5 CONCLUSION AND FUTURE WORK

In this paper, we have presented a novel scene text recognition approach by tuning a vision and language reconstructor to a text recognizer. Our approach, based on mask and reconstruction, not only learns rich visual and semantic representation but also ensures consistency in pre-training and fine-tuning stages. Benefiting from the architecture and innovative modules, our model achieves state-of-the-art performance on standard STR, particularly demonstrating significant improvement in challenging scenarios. Moving forward, there are two potential directions for future expansion: developing a multilingual variant of the proposed method and extending it to line recognition and whole-image recognition.

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
