# OpenReview forum: "VL-Reader: Vision and Language Reconstructor is an Effective Scene Text Recognizer"
_acmmm.org/ACMMM/2024/Conference — MM2024 Poster_

### Official Review · Reviewer_Mq4t · 2024-05-24

**Rating:** 4
**Confidence:** 2

**Summary:**

This work propose a VLM-based text Recognizer where a novel masked visual-linguistic reconstruction is used to jointly learns representations of the vision and semantics of text images.
Experiments on extensive benchmarks show that the proposed VL-Reader outperforms the existing methods by a significant margin, and the analysis reveal how this method advance.

**Strengths:**

1. The proposed method masked visual-linguistic reconstruction is very concise and highly effective.

2. This idea may serves similar V-L tasks intuitively,  providing good insights for multimodal researching.

3. Good writing and clear presentation.

**Limitations:**

I do not find major weakness for this work, except a minor concern: "The average performance of this task has been pushed to a very high level (near 100%) and the shown increase is relatively small." I think adding the significance analysis will help.

I will give a border score first and may rise it after considering other comments in the following discussing period.

**Suitability:**

2

---

### Official Review · Reviewer_fuEH · 2024-05-26

**Rating:** 4
**Confidence:** 3

**Summary:**

This paper proposes a VL-Reader method for scene text recognition. The core of VL-Reader is a Masked Visual-Linguistic Reconstruction (MVLR) loss, which simultaneously reconstructs visual and linguistic information. Then, a Masked Visual-Linguistic Decoder (MVLD) is introduced to further leverage masked vision-language context and achieve bi-modal feature interaction. The training strategy and architecture maintain consistency from pre-training to fine-tuning. Comprehensive experiments are conducted on 6 datasets.

**Strengths:**

1. This paper shows comprehensive experiments on the effectiveness of the proposed MVLR, the visual masking ratio and linguistic masking ratio, model size, the qualitative results, and the reconstruction results.
2. The reconstruction is conducted for vision as well as the text, which is good for vision language information integration. MVLR can inherit the entire architecture across different training stages, rather than only use a vision encoder.
3. The writing and representation are nice.

**Limitations:**

1. This paper lacks comparison with more newly-published work, such as:
(1) “Image as a Language: Revisiting Scene Text Recognition via Balanced, Unified and Synchronized Vision-Language Reasoning Network” in AAAI2024; (2) “Self-supervised Character-to-Character Distillation for Text Recognition” in ICCV2023; (3) “Relational Contrastive Learning for Scene Text Recognition” in ACM MM2023, etc.
2. In equation (1), why the visible patch and masked pathed is processed in different ways? Is there any referenced work or experiments?
3. The vision reconstruction is conducted after MVLD. In the ablation study, the comparison between this strategy and those with reconstruction only after the vision encoder should be presented.
4. In line 512, GPU card should be specified.
5. In Table 1, with synthetic training data, [14][28][47] outperforms the proposed method on specific datasets. Explanations should be provided.
6. Comparison or discussion with LLM-based methods is suggested.

**Suitability:**

3

---

### Official Review · Reviewer_V4nV · 2024-05-31

**Rating:** 4
**Confidence:** 3

**Summary:**

In this study, a vision and language reconstructor model VL-Reader is proposed to recognize scene texts, where two-step (masked and non-masked) training strategy is designed to ecnode and decode paris of image and text, and cross-modal masked VL decoder is introduced to enhance cross-modal representation learning. Experimental result on 6 standard benchmarks and 7 challenging benchmarks have showed the superiority of the proposed model VL-Reader. The proposed model has achieved the best accuracy compared with the other methods. But the model's size seems to much be larger than other models like ABINet. This makes the model be hard to be used on edge devices.

**Strengths:**

The manuscript is well writen.The proposed VL-Reader has achieved the best performance among more than 20 baselines.

**Limitations:**

As for model size analysis, only one model ABINet is compared. More models and their inference speeds (FLOPs) had better be compared. Although the propsed method has achieved good performance, its model parameter seems large.

**Suitability:**

3

---

### Meta-Review · Area_Chair_p13o · 2024-07-02

**Recommendation:** Accept (Poster)
**Confidence:** 4

**Metareview:**

The paper proposes a scene text recognition approach called VL-Reader based on vision and language reconstruction. Firstly, a Masked Visual-Linguistic Reconstruction (MVLR) objective to simultaneously model visual and linguistic information. Then, a Masked Visual-Linguistic Decoder (MVLD) is used to further leverage masked vision-language context and achieve bi-modal feature interaction. The architecture of VL-Reader maintains consistency from pre-training to fine-tuning. In the pretraining stage, VL-Reader reconstructs both masked visual and text tokens, while in the fine-tuning stage, the network degrades to reconstruct all characters from an image without any masked regions.

Strengths:
- The paper is very well-written and clearly explains the motivation and effectiveness of each component. A detailed analysis and evaluation has been conducted and thoroughly described to explain how and why this is approach is superior to the prior ones.
- Every claim made in the paper has been well substantiated by theory as well as results.

Any of the concerns mentioned in the weakness by the reviewers have been well addressed in the rebuttal.